# Training understanding of reversible sentences: a study comparing language-impaired children with age-matched and grammar-matched controls

Hsinjen Julie Hsu[1] and Dorothy V.M. Bishop

Department of Experimental Psychology, University of Oxford, United Kingdom
[1] Current affiliation: Graduate Institute of Audiology and Speech Therapy, National Kaohsiung Normal University, Taiwan

Corresponding author
Dorothy V.M. Bishop,
dorothy.bishop@psy.ox.ac.uk

## ABSTRACT

**Introduction**. Many children with specific language impairment (SLI) have problems with language comprehension, and little is known about how to remediate these. We focused here on errors in interpreting sentences such as "the ball is above the cup", where the spatial configuration depends on word order. We asked whether comprehension of such short reversible sentences could be improved by computerized training, and whether learning by children with SLI resembled that of younger, typically-developing children.

**Methods.** We trained 28 children with SLI aged 6–11 years, 28 typically-developing children aged from 4 to 7 years who were matched to the SLI group for raw scores on a test of receptive grammar, and 20 typically-developing children who were matched to the SLI group on chronological age. A further 20 children with SLI were given pre- and post-test assessments, but did not undergo training. Those in the trained groups were given training on four days using a computer game adopting an errorless learning procedure, during which they had to select pictures to correspond to spoken sentences such as "the cup is above the drum" or "the bird is below the hat". Half the trained children heard sentences using above/below and the other half heard sentences using before/after (with a spatial interpretation). A total of 96 sentences was presented over four sessions. Half the sentences were unique, whereas the remainder consisted of 12 repetitions of each of four sentences that became increasingly familiar as training proceeded.

**Results.** Age-matched control children performed near ceiling ($\geq$90% correct) in the first session and were excluded from the analysis. Around half the trained SLI children also performed this well. Training effects were examined in 15 SLI and 16 grammar-matched children who scored less than 90% correct on the initial training session. Overall, children's scores improved with training. Memory span was a significant predictor of improvement, even after taking into account performance on training session 1. Unlike the grammar-matched controls, children with SLI showed greater accuracy with repeated sentences compared with unique sentences. Training did not improve children's performance on a standardized test of receptive grammar.

**Discussion.** Overall, these results indicate that a subset of children with SLI perform well below ceiling on reversible sentences with three key words and simple syntactic

structure. For these children, weak verbal short-term memory appears to impair comprehension of spoken sentences. In contrast to the general finding that rule-learning benefits from variable input, these children seem to do best if given repeated exposure to the same nouns used with a given sentence frame. Generalisation to other sentences using the same syntactic frame may be more effective if preceded by such item-specific learning.

## INTRODUCTION

For children with specific language impairment (SLI), comprehension of reversible sentences is often a source of particular difficulty (*Bishop, 1979*; *Van der Lely & Dewart, 1986*), but the reasons for this are not fully understood. Theoretical accounts range from those that attribute comprehension impairment to auditory-perceptual problems, to those postulating more specific difficulties in mastering syntactic rules, or in carrying out complex syntactic computations (*Bishop, 1997*).

Most research investigating this phenomenon has focussed on reversible active and passive constructions, testing children's ability to comprehend sentences such as "the boy chases the dog", "the dog chases the boy" or "the dog is chased by the boy". Young children may make errors when asked to act out such sentences because they adopt simple strategies such as "pick up the first-named object and do something with it"—a strategy that is usually effective for active sentences but not for passive ones (*Van der Lely & Dewart, 1986*). As children grow older, errors are more sporadic and less easy to explain in terms of a strategic account. For instance, *Bishop (1982)* used a multiple choice test, where the correct picture had to be selected from an array that included foils depicting the reversed agent-patient relationship. Although some children systematically misinterpreted all passives, these were unusual cases with severe comprehension problems associated with acquired epileptic aphasia. Other children with more typical language impairment performed below age-level, but their errors were less systematic.

Poor comprehension of grammatically complex sentences has been recognised as a problem for many language-impaired children and a handful of experimental interventions have been developed to address these (*Ebbels, 2007*; *Ebbels & Van der Lely, 2001*; *Ebbels et al., 2014*; *Levy & Friedmann, 2009*; *Riches, 2013*). To help children overcome comprehension problems, we need to understand the reasons for their difficulties with these types of construction. One possibility is that perceptual difficulties are implicated. For the child who has difficulty perceiving brief, non-salient morphemes, "the boy is chasing the dog" may be hard to distinguish from "the boy is chased by the dog", leading to poor performance on a multiple-choice comprehension task. A second possibility is that perception is adequate but the complex syntax of active and passive sentences

is the stumbling block. Various theories of SLI have proposed difficulties either with specific areas of syntax (*Rice, Wexler & Cleave, 1995*), or more generally with the complex computations that are involved in establishing grammatical relationships between different sentence components (*Van der Lely, 1996*; *Van der Lely, 2005*). Such accounts cannot explain, though, why children should have difficulty with even simple reversible constructions, namely those using a spatial preposition, such as "the comb is above the flower".

Like other open-class words, spatial prepositions are very common in the language. In a rank-ordered list of words in the British National Corpus, the prepositions "before", "after", "above" and "below" occur at ranks 185, 111, 786 and 1656 respectively. These frequencies are similar to common concrete nouns such as "house" (rank 191), "cup" (rank 831) or "cat" (rank 1758). Children appear to master the meanings of spatial prepositions, however, relatively late (*Durkin, 1981*), especially if contextual cues are removed. In many situations, the meaning of a spatial preposition can be disambiguated by context: for instance, we can put a key in a cup, but we cannot put a cup in a key, and even young children will do well when asked to act out an instruction such as "put the key in the cup" (*Wilcox & Palermo, 1975*). When such cues to meaning are eliminated, however, understanding of spatial prepositions is much harder. For instance, the Test for Reception of Grammar-2 (*Bishop, 2003*) includes four multiple choice items, each of which requires a sentence such as "the flower is above the cup" to be matched to the correct picture from an array. The array includes three foils depicting different spatial arrangements, including one corresponding to the reversed sentence. Not until nine years of age do 90% of typical children show reliable mastery of this construction.

*Bishop, Adams & Rosen (2006)* conducted a training study that confirmed the difficulty that many language-impaired children had with simple reversible sentences, even when perceptual problems were unlikely to play a part because all the items were acoustically distinct, familiar and shown pictorially. As with the earlier study with active and passive sentences (*Bishop, 1982*), poor performance did not indicate a total failure to understand the sentences; scores were well above the level expected if the children had just been guessing. The problem, rather, was a lack of automaticity in comprehending these sentences, with performance being slow, effortful and error-prone.

This pattern of performance suggested two alternative explanations for children's comprehension difficulties. One possibility is that the errors arise because of problems maintaining a sentence in short-term memory while computing its meaning. Verbal short-term memory is usually impaired in SLI, and it has been shown to play a role in comprehension problems (*Montgomery, 1995*; *Montgomery, 2000*; *Montgomery, Magimairaj & Finney, 2010*). However, the literature on this topic has focused mainly on syntactically complex sentences; the question is whether poor memory could limit comprehension for simple prepositional sentences comprising only three key elements (e.g., $X$ is below $Y$), presented with pictorial support.

Another possibility is that there is difficulty learning concepts that express relationship between other items. For instance, the prepositions "above" and "below" express a

spatial relationship between two arguments, *X* and *Y* (e.g., *X* is above *Y*; *X* is below *Y*). To learn the meaning of the "_ *is above* _"and"_ *is below* _" constructions, the child must perceive the common spatial relationship that is present when they are used and attach meaning to the invariant part (e.g., "*is above*" or "*is below*"), so that a general, abstract meaning is understood, regardless of the identity of *X* and *Y*. Previous studies with typically-developing individuals have shown that high variability facilitates detection of invariant parts of sentences (*Gómez, 2002*). Similar results were found with typically-developing children in an artificial-grammar learning study, where syntactic constructions were learned better in a high variability condition than in a low variability condition (*Hsu, Tomblin & Christiansen, 2014*). However, the opposite pattern was found for age-matched children with SLI, who did better in the low variability condition. This result fits with a suggestion that children with SLI may have difficulty learning abstract meanings in syntactic patterns, and might rely instead on rote learning of whole sentences (*Hsu & Bishop, 2010*). Thus they might learn the specific meaning of a phrase containing a preposition as a whole entity (e.g., "the room above the stairs") without generalising the meaning of the preposition.

The current study was designed to explore problems that children with SLI have in understanding reversible sentences. It used a training task modelled on that used by *Bishop, Adams & Rosen (2006)* but restricted to simple reversible sentences containing spatial prepositions: either above/below or before/after (with a spatial interpretation). These constructions were selected to correspond to the simplest type of sentence where word order can disambiguate meaning. Children were trained to respond to such sentences in a game-like task using an errorless learning format. Three key questions were addressed in this study:

1. Do children with SLI have trouble learning to respond accurately to reversible sentences compared with their age-matched controls, even when a very simple syntactic frame is used, and training focuses on just one meaning contrast?
2. Does variability of nouns used in training sentences help or hinder comprehension of children with SLI? This question was addressed by comparing performance on a set of unique sentences, each of which used different nouns, with a set of sentences using the same nouns, presented repeatedly throughout training. The performance of children with SLI was compared with that of younger typically-developing children matched on overall comprehension level.
3. Does comprehension of reversible sentences depend on capacity of verbal short-term memory?

## METHODS

### Ethics approval

Approval for this study was given by the University of Oxford Medical Sciences Division Research Ethics Committee, approval reference MSD/IDREC/2009/28. Parents of all participants gave written informed consent, and the children gave assent after the study was explained in age-appropriate language.

## Participants

A total of 96 children took part in the study, subdivided into four groups: (a) 6 to 11 year-old children with SLI who received language training (SLI-T, $N = 28$); (b) typically-developing children aged from 4 to 7 years, who were matched for raw scores on a test of receptive grammar (grammar-matched controls, $N = 28$); (c) typically-developing children matched for chronological age (age-matched controls, $N = 20$); and (d) 7 to 11 year-old children with SLI who did not receive language training (SLI-U, $N = 20$). This is the same sample whose procedural learning skills were documented by *Hsu & Bishop (2014)*. This report focuses on groups (a) and (b), i.e., the SLI and grammar-matched groups who underwent training. Data from the untrained children with SLI (d) are of interest solely for identifying whether there are beneficial effects of training that transfer to other tasks, in which case the trained SLI group should outperform the untrained SLI group on a post-test administered after training. We anticipated that the age-matched control group (c) would find the reversible sentences comprehension task very easy, but they were included to test this assumption.

The children with SLI were recruited from special schools for children with language impairment or support units in mainstream schools. Children were included if they met all of the following criteria:

(1) Performed at least 1 SD below the normative mean on at least two out of the following six standardized tests: the British Picture Vocabulary Scales II (BPVS II) (*Dunn et al., 1997*), Test for Reception of Grammar-Electronic (TROG-E) (*Bishop, 2005*), the comprehension subtest of the Expression, Reception and Recall of Narrative Instrument (ERRNI), (*Bishop, 2004*), repetition of nonsense words from the Developmental Neuropsychological Assessment (NEPSY) (*Korkman, Kirk & Kemp, 1998*), and syntactic formulation and naming subtests of the Assessment of Comprehension and Expression 6–11 (ACE 6–11) (*Adams et al., 2001*).

(2) Had nonverbal ability within the normal range (no more than 1 SD below average), as measured with Raven's Coloured Progressive Matrices (*Raven, Court & Raven, 1986*).

(3) Were able to hear a pure tone of 20 dB or less in the better ear, at 500, 1,000, 2,000 and 4,000 Hz.

(4) Had English as their native language.

(5) Did not have a diagnosis of other neurodevelopmental disorders such as autism or Down Syndrome.

Children meeting the inclusion criteria for SLI were randomly assigned to either SLI-T (i.e., trained group) or SLI-U (i.e., untrained group). These two SLI groups did not differ in age, nonverbal IQ, or any of the standardized language tests.

The same test battery was used to confirm language status for children in the two control groups. Both these groups met the same criteria for nonverbal IQ, hearing and native language, and did not have a history of speech, language, social or psychological impairments. At least five of six standardized language test scores were within normal

Table 1 **Mean (SD) age and test scores for four groups.** Means with different superscripts differ significantly from one another on post hoc Sidak test, $p < .05$.

| | SLI-Trained; $N = 28$ | SLI-Untrained; $N = 20$ | Grammar-matched; $N = 28$ | Age-matched $N = 20$ |
|---|---|---|---|---|
| Age (yr) | 8.6 (1.32) | 9.1 (1.32) | 5.8 (0.86) | 8.9 (0.77) |
| RCPM SS | 102.9 (13.25) | 100.6 (10.38) | 105.2 (8.47) | 105.8 (11.35) |
| TROG-E raw blocks | 8.3 (4.00)[a] | 9.7 (3.34)[a] | 9.8 (3.11)[a] | 14.8 (2.38)[b] |
| TROG-E SS | 73.3 (14.29)[a] | 75.7 (10.64)[a] | 102.3 (13.96)[b] | 97.8 (10.31)[b] |
| BPVSII raw | 69.5 (17.77)[a] | 75.4 (14.68)[a] | 67.1 (12.95)[a] | 92.6 (9.16)[b] |
| BPVSII SS | 87.8 (13.18)[a] | 87.7 (8.89)[a] | 108.3 (9.51)[b] | 102.6 (7.58)[b] |
| NEPSY nonwords raw | 22.3 (8.90)[a] | 23.9 (8.39)[a] | 27.0 (7.60) | 32.2 (8.48)[b] |
| NEPSY nonwords SS | 84.1 (16.78)[a] | 85.5 (14.13)[a] | 104.3 (15.01)[b] | 101.8 (15.75)[b] |
| ERRNI Comprehension raw | 8.3 (3.25)[a] | 9.7 (3.96)[a] | 8.4 (3.12)[a] | 12.9 (2.56)[b] |
| ERRNI Comprehension SS | 82.5 (15.17)[a] | 87.9 (16.29)[a] | 103.0 (14.6)[b] | 101.9 (12.58)[b] |
| ACE Naming raw | 10.2 (4.32)[a] | 12.3 (2.94)[a] | 10.0 (2.84)[a] | 16.3 (3.01)[b] |
| ACE Naming SS | 81.6 (13.2)[a] | 83.8 (9.16)[a] | 101.1 (7.98)[b] | 98.8 (10.99)[b] |
| ACE Syntax raw | 15.0 (5.16)[a] | 17.2 (6.64)[a] | 17.8 (5.93)[a] | 24.1 (5.13)[b] |
| ACE Syntax SS | 80.7 (9.2)[a] | 83.3 (13.7)[a] | 105.5 (14.16)[b] | 98.5 (15.05)[b] |
| $N$ impaired tests | 3.3 (1.38)[a] | 3.2 (1.15)[a] | 0.3 (0.48)[b] | 0.4 (0.5)[b] |
| Word Span | 4.0 (1.07)[a] | 4.1 (0.91)[a] | 4.1 (0.88)[a] | 4.9 (0.79)[b] |

limits (no more than 1 SD below average). Descriptive information on the participants is given in Table 1.

The children in the grammar-matched group were aged between 4 and 7 years and were matched as closely as possible with the children in the SLI-T group on receptive grammar using TROG-E, a standardized test that contains four four-choice test items for each of twenty different syntactic contrasts (e.g., negation, subject relative clause, post-modified subject, etc.), including one block that tests comprehension of reversible above/below sentences. A block is scored as passed if all four items in that block are correct. Each child in the grammar-matched group had a TROG-E raw score within 3 blocks of one of the children in the SLI-T group.

### Sentence comprehension training

The computerized training program was based on that used by *Bishop, Adams & Rosen (2006)*. Children received 4 training sessions, with one session per day over a period of 4–6 days. They spent around 5–7 min per day on the sentence comprehension program.

In each session, children were presented with 24 sentences containing either *before/after* or *above/below*. Among the 24 sentences, there were 12 sentences that occurred only once (unique items) and four sentences that appeared three times across a training session (repeated items). The same repeated items were used in each session (see Supplemental Information). Thus, in total, children heard 48 unique sentences and four sentences that were repeated 12 times across the four training sessions. This manipulation allowed us to consider how sentence variability affected learning in children with SLI.

Children were randomly assigned to be trained with reversible sentences containing either *before/after* or *above/below*. Those assigned to the above/below condition listened to 12 sentences containing the preposition "*above*" and 12 sentences containing the preposition "*below*" in each training session. For each sentence, they selected two target pictures from an array of four pictures and moved each picture to an upper or a lower chamber of a space shuttle, to match the meaning of the sentence (Fig. 1). Children assigned to the before/after condition followed the same test procedure as in the above/below condition, but were asked to move pictures to the front and back carriages of a train to indicate the spatial relationship of the two target pictures.

A game-like format using an errorless learning procedure was adopted. If the child made a correct response, there was a visible reward, with a cartoon character heading a football over a bar, to add to the child's collection. If a wrong response was made, the child had an opportunity to try again. The tester would say "Do you want to try again? Or you can click on the Help button here". The Help button provided a visual cue to indicate which items needed to be selected. To minimize errors due to misperception or forgetting, a Talk button was included: the tester explained to the child that when this button was pressed, the test sentence would be spoken again.

One point was given for each trial when a correct answer was provided on the first attempt by the child, even if the child had to click on the Talk button to listen to the target sentence again. No points were given if the target picture was selected after more than one attempt by the child or if the Help button was used. The program automatically recorded timing of responses in milliseconds. Although times recorded on a laptop computer are not very accurate, and likely to vary from one machine to another (*Plant & Quinlan, 2013*), we previously found that this measure was sensitive to the large within-subject changes in speed associated with training (*Bishop, Adams & Rosen, 2006*). For each session, the child's median response time for correct responses was taken separately for the repeated and unique item sets.

One to three days after training session 4, children were seen for another session (session 5) in which the same computerized sentence comprehension task was conducted, but using the set of prepositions (i.e., *above/below* or *before/after*) on which children had not been trained. This made it possible to check whether any improvement was specific to the trained preposition, or whether it generalized to a different pair of prepositions.

## Memory measures

### Nonword repetition

Raw score on the NEPSY repetition of nonsense words (*Korkman, Kirk & Kemp, 1998*) was used as a measure of phonological short-term memory. In this test, children listen to 13 recorded spoken nonwords through headphones and must repeat what they hear. The nonwords range in length from 2 to 5 syllables, and one point is awarded for each syllable correctly spoken. N.B. This test was also used as one of the six language measures that could contribute to identification of SLI.

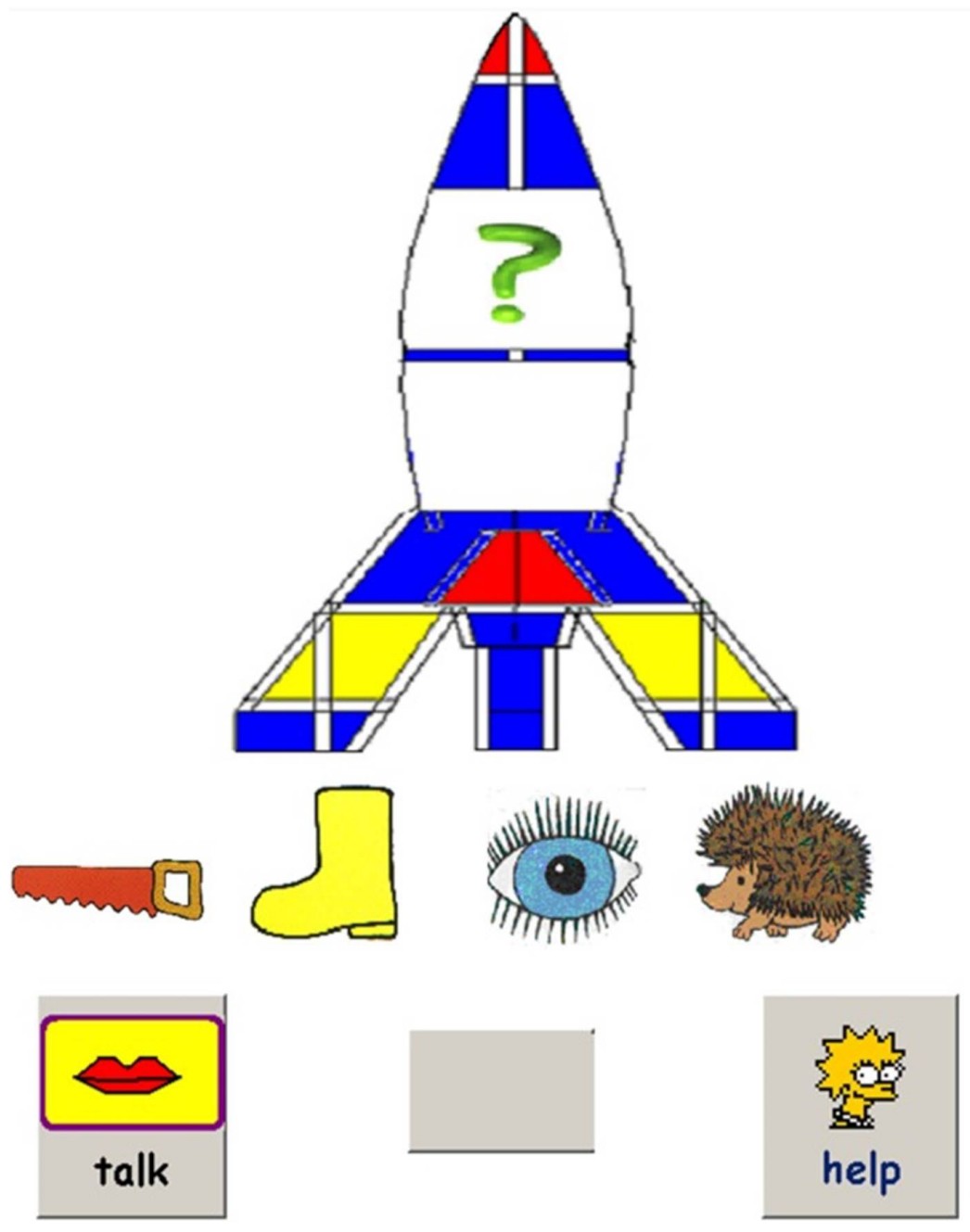

**Figure 1  Sample screenshot from comprehension training program.** The child hears a sentence such as "the hedgehog is above the boot", and must click first on the hedgehog and then on the boot to move them into correct positions in the rocket. The question mark denotes the next position to be filled. The child can press "talk" to have the sentence repeated, and can press "help" to have visual cues added to denote which items must be moved. The format for the preposition pair "before/after" was identical except that a train was shown, and the task was to select items to fill trucks of the train in the correct horizontal order.

| Table 2 | Schedule of assessment and training. | |
|---|---|---|
| **Session** | **Groups** | **Activity** |
| 1 | All | TROG-E, BPVS-II, Raven's Matrices, hearing screen |
| 2 | All | Other language tests (see Table 1) |
| 3 | All except SLI-U | Training session 1 |
| 4 | All except SLI-U | Training session 2 |
| 5 | All except SLI-U | Training session 3 |
| 6 | All except SLI-U | Training session 4 |
| 7 | All except SLI-U | Transfer of training, session 5 (untrained preposition) |
| 7 | All | Posttest: BPVS-II and TROG-E (parallel form) |

### Word span

A computerized word span task was used, in which the child's task was to select pictures to be placed in a horizontal line of numbered fishing nets in the correct order. Children listened to lists of familiar words and saw pictures of the words appear on the computer screen immediately after the auditory presentation of the words. They then clicked the pictures in the same order as they had heard the corresponding words. Once the first picture was clicked, it moved automatically to the first fishing net, and so on for the rest of the pictures until all the pictures were clicked. Items started with a list length of three, and increased by one item each time the list was recalled correctly in the right order. When an incorrect response was given, a second attempt at the same list length was provided. If both trials were failed at list length three, a list length of two was presented. The program stopped automatically when two successive trials were failed at a given list level. The dependent measure was word span, i.e., the longest list length correctly reproduced at least once. The word lists were composed of monosyllabic nouns that were found in the lexicon of two-year-old typically-developing children (*Hamilton, Plunkett & Schafer, 2000*).

### Testing schedule

All children, except those assigned to the SLI-U group, were seen for two weeks during which they completed two screening sessions (assessing language, hearing, nonverbal IQ), followed by four sessions of language training and a post-test session (see Table 2). Children in the SLI-U group were seen only for the first two screening sessions and the post-test session. On the training days children were trained on three computerised tasks, including the computerized sentence comprehension task. The other tasks involved training vocabulary and nonverbal paired-associate learning, and will be described elsewhere. One to two days after the completion of training, a post-test session was given to all children, including the SLI-U group. In the post-test receptive grammar was assessed using a different parallel form of TROG-E.

## RESULTS

It would not be possible to demonstrate improvement with training for children who were highly accurate in session 1. We therefore restricted analysis of training effects to those who performed below ceiling (i.e., less than 90% correct) with reversible sentences on session 1. We had anticipated that age-matched controls would find this task very easy, and this proved to be the case, with only one child scoring below ceiling. Accordingly the age-matched group was excluded from further consideration. Of the 28 children in the SLI group, 15 (53%) scored below 90% in the first training session, as did 16 of 28 (57%) children in the grammar-matched control group. It is also of interest to consider how many children score at chance—indicating no understanding of the contrast. Since virtually all errors consisted of selecting the correct items in the wrong order, this can effectively be treated as a two-choice task, so a score of between 8–15 out of 24 correct can be regarded as chance level performance (binomial theorem, $p < .05$). Nine of the children in the SLI group and four of those in the grammar-matched group scored at chance on session 1. Six children from the SLI group and twelve from the grammar-matched group scored above chance but below ceiling.

Analysis proceeded in four stages: first, exploratory comparisons were made between the children who did and did not score close to ceiling in session 1, to identify characteristics of those with adequate comprehension. Second, for those who scored below ceiling, we compared rates of learning for unique and repeated items in the SLI and grammar-matched groups. Third, we examined scores on the post-test to check for generalization of training effects. Finally, we conducted further analyses to examine memory predictors of learning.

### Comparison of background measures for children who performed above or below ceiling in session 1

Table 3 shows the mean age and test scores for the SLI and grammar-matched groups subdivided into those who did and did not score at ceiling on training session 1. A two-way MANOVA was conducted on raw language scores with comprehension status (above-ceiling vs. below-ceiling) as one factor, and group (SLI-T vs. grammar-matched control) as the other. Those who scored below ceiling tended to be younger, $F(1, 52) = 4.41, p = .041$. There were just two language measures that were significantly different for the above-ceiling and below-ceiling subgroups: NEPSY repetition of nonsense words, $F(1, 52) = 5.14, p = .027$, and ACE Syntactic Formulation, $F(1, 52) = 4.71, p = .034$). These same two measures also differed significantly between the SLI and grammar-matched groups, NEPSY repetition of nonsense words: $F(1, 52) = 5.15, p = .027$; ACE Syntactic Formulation, $F(1, 52) = 4.30, p = .043$). Thus, although we had attempted to match the SLI and grammar-matched groups on receptive syntax, the SLI children scored more poorly on a measure of phonological short-term memory and on syntactic formulation, both of which were measures associated with poor performance on the comprehension training task. There were no significant interactions between group and ceiling status: i.e., the pattern of differences for those above and below ceiling was similar in the SLI and grammar-matched groups.

Table 3 **Mean (SD) age and test scores.** SLI-T and grammar-matched groups subdivided according to whether above or below ceiling in Session 1. Means with different superscripts differ significantly from one another on post hoc Sidak test, $p < .05$.

| | SLI-T, below ceiling, $N = 15$ | SLI-T, above ceiling, $N = 13$ | Grammar-matched, below ceiling, $N = 16$ | Grammar-matched, above ceiling, $N = 12$ |
|---|---|---|---|---|
| Age (yr) | 8.4 (1.16) | 8.8 (1.48) | 5.5 (0.78) | 6.2 (0.80) |
| RCPM SS | 100.9 (12.52) | 105.2 (14.18) | 103.4 (8.45) | 107.6 (8.24) |
| TROG-E raw blocks | 7.7 (2.97) | 9.1 (4.96) | 8.9 (3.21) | 10.8 (2.73) |
| TROG-E SS | 70.7 (11.26) | 76.2 (17.13) | 103.3 (13.59) | 101.1 (14.95) |
| BPVSII raw | 70.4 (14.31) | 68.5 (21.66) | 64.1 (12.59) | 71.1 (12.85) |
| BPVSII SS | 90.5 (10.48) | 84.5 (15.54) | 109.3 (9.97) | 106.8 (9.09) |
| NEPSY nonwords raw | 19.9 (7.47) | 25.0 (9.92) | 25.0 (8.59) | 29.8 (5.19) |
| NEPSY nonwords SS | 81.0 (15.26) | 87.7 (18.33) | 100.9 (16.35) | 108.8 (12.27) |
| ERRNI Comprehension raw | 7.7 (3.54) | 8.9 (2.9) | 7.5 (3.18) | 9.6 (2.71) |
| ERRNI Comprehension SS | 82.2 (15.29) | 82.9 (15.64) | 102.5 (13.64) | 103.7 (16.39) |
| ACE Naming raw | 10.0 (4.88) | 10.4 (3.75) | 9.8 (2.67) | 10.4 (3.12) |
| ACE Naming SS | 83.7 (14.08) | 79.2 (12.22) | 102.5 (7.07) | 99.2 (9.00) |
| ACE Syntax raw | 14.0 (5.40) | 16.1 (4.84) | 15.9 (6.27) | 20.2 (4.65) |
| ACE Syntax SS | 80.7 (9.61) | 80.8 (9.09) | 102.8 (13.90) | 109.2 (14.28) |
| $N$ impaired tests | 3.1 (1.46) | 3.4 (1.33) | 0.4 (0.50) | 0.3 (0.45) |
| Word Span | 3.7 (1.05) | 4.3 (1.03) | 4.0 (0.89) | 4.3 (0.87) |

## Performance on unique and repeated sentences for children not at ceiling

Figures 2A and 2B show the mean accuracy across the four training days for the SLI-T and grammar-matched children who scored below ceiling.[1] (Data from the post-test, shown as session 5, will be discussed later).

Data were analysed using a mixed-design ANOVA, with session (1 to 4) and item type (unique vs. repeated) as repeated measures, and group as between-subjects factor. One potential problem with multiway analysis of variance is that the probability of a Type 1 error is increased because separate null hypotheses are tested (e.g., two main effects and one interaction in a 2-way design) (*Cramer et al., 2014*). To counteract this, a correction for the alpha level based on the number of null hypotheses tested should be implemented. Accordingly, to correct for experiment-wise error rate, we adopted a significance level of $p = .05/7 = .007$, adjusting for the three main effects and four interactions. Effects that have associated $p < .05$ but $> .007$ are described as marginal.

Looking at main effects first, the overall effect of group was not significant: $F(1, 29) = 0.97, p = .332$. The effect of item type (unique vs. repeated) was not significant, $F(1, 29) = 3.20, p = .084$. The effect of session was significant, $F(3, 87) = 10.96, p < .001$, partial $\eta^2 = .274$. Planned contrasts showed a linear effect of session accounted for a high proportion of variance, indicating learning: $F(1, 29) = 23.6, p < .001$, partial $\eta^2 = .449$.

[1] At the suggestion of a reviewer, we also divided the SLI-T children according to whether or not they scored at chance on session 1. We found that the learning profiles of these two subgroups are very similar to that based on whole group data.

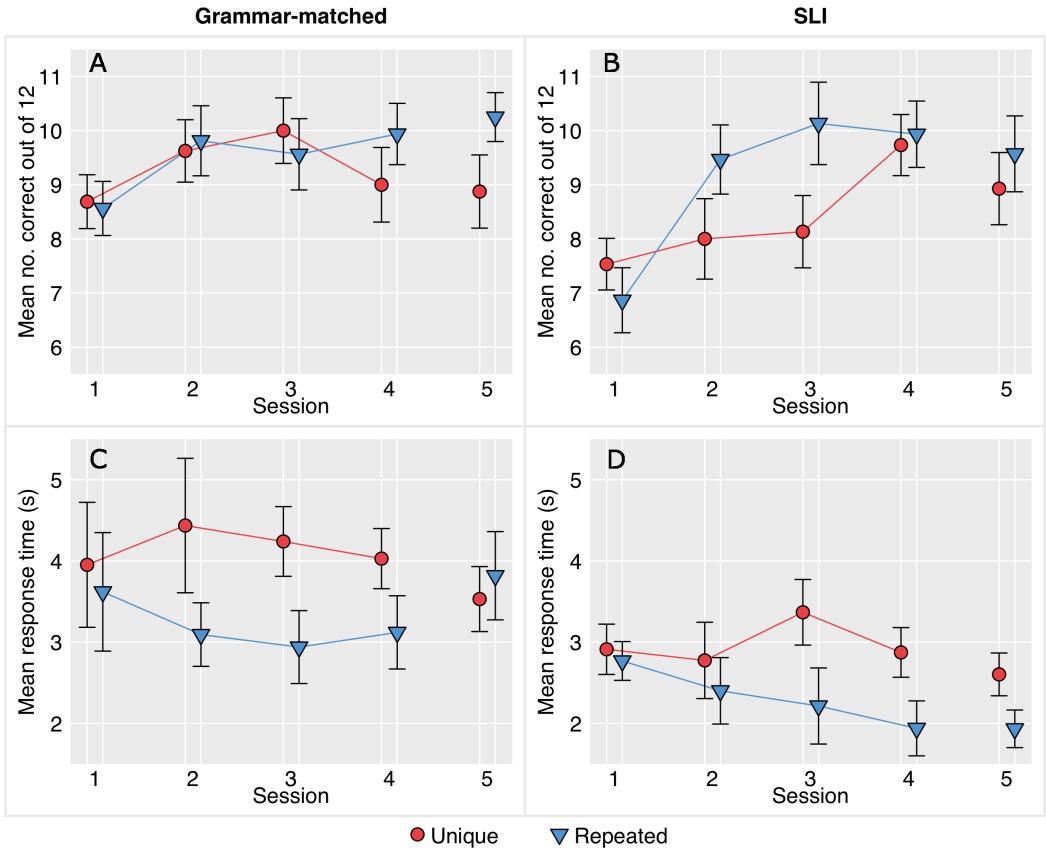

**Figure 2 Mean items correct and response times across four training sessions and a post-test session (5) for children who scored below 90% correct on session 1.** (A) and (B) show mean items correct, and (C) and (D) show response times. Grammar-matched children in (A) and (C) and SLI-T in (B) and (D). Error bars show standard errors.

We next considered interactions of group with the within-subjects factors. We were specifically interested in the interaction between item type and group, which was non-significant, $F(1, 29) = 1.49, p = .231$. The interaction between session and group was marginal, $F(3, 87) = 2.83, p = .043$. However, the three way interaction between group, item type and session was significant, $F(3, 87) = 5.44, p = .002$, partial $\eta^2 = .158$. To explore this further, a two-way ANOVA between item type and session was performed separately for each group. For the grammar-matched controls, neither main effect or interaction reached significance, though there was a trend for an effect of session: $F(3, 45) = 3.85, p = .016$. For the SLI-T group, the main effect of item type was non-significant, but there was a significant effect of session: $F(3, 42) = 9.19, p < .001$, and a significant interaction between session and item type: $F(3, 42) = 6.38, p = .001$.

Finally, the difference between unique and repeated items was compared for each group at each session using a matched pairs $t$-test with $p$ set to .05. For the SLI-T group, the superiority of repeated over unique items was significant for session 2, $t(14) = 3.29, p = .005$, and session 3, $t(14) = 2.56, p = .023$, but not for the other sessions. For the grammar-matched group, the difference between item types was significant

only for session 4, $t(15) = 2.46, p = .027$, but as can be seen from Fig. 2, this reflected deterioration in performance on unique items relative to session 3, rather than superior learning of repeated items.

Data on response times are shown in Figs. 2C and 2D. A test of homogeneity of variances indicated significant differences in covariance matrices between groups ($p = .046$), so data were log-transformed. This led to a non-significant test for homogeneity ($p = .621$), but the session factor was significant on Mauchly's test of sphericity (Mauchly's $w = .452, d.f. = 5, p = .001$), so Greenhouse-Geisser correction was applied.

The overall effect of group was marginal, reflecting a trend for faster responses by the SLI group compared to grammar-matched controls, $F(1, 29) = 4.46, p = .043, partial\ \eta^2 = .133$. Item type exerted a substantial effect on response times, with faster responses to repeated items, $F(1, 29) = 75.4, p < .001$, partial $\eta^2 = .722$, and there was also a significant interaction between item type and session, $F(3, 87) = 10.8, p < .001$, partial $\eta^2 = .271$. However, there was no interaction between group and session or item type. Correct responses to repeated items became increasingly rapid as children became more familiar with them, but this effect was equivalent in children with SLI and their grammar-matched controls.

## Post-test measures of generalization

### Performance on session 5

Data gathered after the training allowed us to see whether there had been any generalization from the training. The first source of data was from session 5, in which children were given the same training game, but with the preposition pair they had not been trained on (above-below or before-after). Although the prepositions were new, half the items contained the noun sequences that had previously been used in repeated items. Data were missing for this post-test for one child in the SLI-T group. Means and standard errors for accuracy and response times are shown in Fig. 2 (session 5).

The expectation is that performance on a new preposition pair should be no better than was seen in session 1, unless the child had learned some general strategy for performing this kind of comprehension task. The mean scores for session 5 were higher than that of session 1 for both groups, the mean (both item types combined) increasing from 17.25 (SD = 3.49) to 19.13 (SD = 4.35) in the grammar-matched group and from 14.71 (SD = 3.67) to 18.50 (SD = 4.29) in the SLI-T group. This effect was examined using a mixed-design ANOVA comparing session 1 and session 5 scores for Unique and Repeated items in SLI and grammar-matched groups. The main effect of session was significant, $F(1, 28) = 15.43, p = .001$, partial $\eta^2 = .355$, though neither the main effect of group nor the group by session nor the group x item type interaction was significant. This indicates that training led to a general improvement (i.e, generalization) in preposition comprehension. There was a marginal interaction of session and item type, $F(1, 28) = 6.46, p = .017$, partial $\eta^2 = .188$. The three-way interaction of group x session x item type was not significant.

 

**Table 4 Pre- and post-training TROG-E blocks passed.** Groups subdivided according to initial level of performance on Session 1 (Age-matched, grammar-matched and SLI-T groups) or TROG-E (SLI-U group).

| Group | Initial performance on training game | Trained? | N | Mean (SD) pre | Mean (SD) post | t | P |
|---|---|---|---|---|---|---|---|
| Age-matched control | At ceiling | Yes | 19 | 14.79 (2.44) | 15.32 (2.36) | 0.87 | .399 |
| Grammar-matched control | Below ceiling | Yes | 16 | 8.94 (3.21) | 9.94 (4.02) | 1.71 | .108 |
| Grammar-matched control | At ceiling | Yes | 12 | 10.83 (2.73) | 11.42 (3.68) | 0.71 | .492 |
| SLI-T | Below ceiling | Yes | 15 | 7.67 (2.97) | 7.13 (4.45) | −0.71 | .488 |
| SLI-T | At ceiling | Yes | 13 | 9.08 (4.96) | 9.85 (5.14) | 0.70 | .495 |
| SLI-U | Low TROG-E | No | 8 | 6.54 (2.77) | 7.25 (3.62) | 0.58 | .576 |
| SLI-U | High TROG-E | No | 12 | 11.83 (1.40) | 12.25 (2.95) | 0.57 | .581 |

The marginal interaction between session and item type was unexpected, because all sentences were novel in session 5 (i.e., they used a different preposition). Because of its potential clinical significance, we explored this effect further, using single-sample $t$-tests to compare gain in score with zero. The mean gain for Unique items was $0.70 \pm .491$, which did not differ significantly from zero, $t(29) = 1.42, p = .165$. The mean gain for Repeated items was $2.07 \pm .401$, $t(29) = 5.15, p < .001$. The pattern of results indicated that learning of a new preposition might be facilitated for all children by use of nouns they had recently been trained with.

A parallel analysis was conducted on log response times, comparing session 1 and session 5. There was a marginal overall effect of group, reflecting faster response times by the SLI group, $F(1, 28) = 6.13, p = .020$, partial $\eta^2 = .180$. There was also a significant 3-way interaction between group x session x item type, $F(1, 28) = 9.19, p = .005$, partial $\eta^2 = .247$. This reflected the fact that Unique and Repeated RTs did not differ significantly except for the SLI group in Session 5, whose RT was faster to the Repeated items, which used the same nouns as before, but with a different preposition.

### Pre-test vs. post-test performance on TROG-E

Another way of assessing the impact of training is to consider changes in overall performance on TROG-E, where different parallel forms had been administered before and after the training. Table 4 shows the mean TROG-E raw scores of the age-matched controls, grammar-matched controls, and the two SLI groups from the initial screening and final post-test session. Children were subdivided according to whether or not they were at ceiling on session 1 of the training. For the untrained SLI group we did not have the information to make this distinction, so children were subdivided instead according to initial TROG-E blocks passed (below 10 vs. 10 or more) in order to create comparable subgroups of different levels of comprehension ability. Paired $t$-tests were conducted to determine the difference in mean scores. None of these groups showed a significantly different performance in the post-test in comparison to the initial test, with the SLI-T group showing a (nonsignificant) trend for worsening performance. There was thus no

**Table 5 Score on training session 1 (max 24) predicted by age and memory measures (raw scores).** (A) All children from grammar-matched and SLI groups ($N = 56$). (B) Children from grammar-matched and SLI groups who perform below ceiling in session 1 ($N = 31$).

| Variable | Zero-order $r$ | | | | beta |
|---|---|---|---|---|---|
| | Age (yr) | Nonword repetition | Word span | Session 1 score | |
| **A** | | | | | |
| Age (yr) | | −.20 | .16 | .13 | .16 |
| Nonword rep. | | | .32* | .34* | .29* |
| Word span | | | | .35* | .23 |
| Mean | 7.19 | 24.6 | 4.03 | 19.1 | |
| SD | 1.78 | 8.54 | 0.97 | 4.62 | $R^2 = .20$ |
| **B** | | | | | |
| Age (yr) | | −.29 | .04 | −.13 | −.11 |
| Nonword rep. | | | .27 | .27 | .15 |
| Word span | | | | .35 | .32 |
| Mean | 6.88 | 22.6 | 3.83 | 15.9 | |
| SD | 1.74 | 8.34 | 0.97 | 3.84 | $R^2 = .17$ |

**Notes.**

* $p < .05$.

evidence that improvement on the computerized preposition training task generalized to the children's overall sentence comprehension ability.

To further assess impact of training on TROG-E performance, for children trained on the "above-below" contrast, performance on the four reversible above/below items of TROG-E was compared before and after training. For the eight grammar-matched children who were trained on "above" and "below", the number of correct responses on TROG-E before and after training was 2.75 and 3.25 out of 4 test items, a non-significant improvement ($t_{(7)} = -1.0, p = .35$). For the eleven SLI children who were trained on "above" vs. "below", the performance on the four reversible above/below in TROG-E was identical before and after training, with a mean of 2.27 out of 4 items correct. Thus despite training with feedback on 96 sentences using this contrast, the children showed no indication of having reached stable understanding of above-below.

## Memory measures as predictors of learning on the sentence comprehension task

Two multiple regression analyses were run using the full sample of grammar-matched controls and SLI children. These examined the relationship between comprehension and memory measures. First, age and raw scores on the two memory measures, nonword repetition and word span, were used as predictors of comprehension performance on session 1. Results are shown in Table 5. The analysis was a non-hierarchical multiple regression, so the beta values indicate the extent to which each variable predicts the dependent variable when all other variables have been taken into account. Both of the memory measures correlated significantly with day 1 comprehension to a similar extent; but nonword repetition emerged as the only significant predictor in the full regression model.

**Table 6 Score on training session 4 (out of 24) predicted by age, memory measures (raw scores), and session 1 score.** (A) All children from grammar-matched and SLI groups ($N = 56$). (B) Children from grammar-matched and SLI groups who perform below ceiling in session 1 ($N = 31$).

| Variable | Zero-order $r$ | | | | beta |
|---|---|---|---|---|---|
| | Nonword repetition | Word span | Session 1 score | Session 4 score | |
| **A** | | | | | |
| Age (yr) | −.20 | .16 | .14 | .19 | .08 |
| Nonword rep. | | .32* | .33* | .16 | −.06 |
| Word span | | | .35** | .45** | .33* |
| Score day 1 | | | | .46** | .35** |
| | | | | | $R^2 = .31$ |
| **B** | | | | | |
| Age (yr) | −.29 | .04 | −.14 | .21 | .24 |
| Nonword rep. | | .27 | .27 | 0 | −.14 |
| Word span | | | .35 | .43* | .34 |
| Score day 1 | | | | .42* | .38* |
| | | | | | $R^2 = .34$ |

Notes.
* $p < .05$.
** $p < .01$.

Using the full sample has the benefit of enhancing statistical power, but the disadvantage that results are somewhat skewed by inclusion of cases who scored close to ceiling levels. The analysis was therefore repeated just using the 31 children scoring below 90% on session 1 (Table 5B). The pattern of results was similar, but correlations of memory measures with session 1 score were no longer statistically significant with this smaller sample size.

Next, number of items correct on the last day of training (session 4) was taken as the dependent measure, with number correct in session 1, age, nonword repetition and word span as predictors. This analysis was run both with all 56 children (including those who performed near ceiling), and with the smaller sample of 31 children who performed below ceiling on session 1.

Results for the full sample are shown in Table 6A. The analysis showed that memory span predicted amount learned (i.e., session 4 total after adjusting for session1 total), whereas nonword repetition did not. The pattern of results was the same when analysis was confined to the sample of children who scored below ceiling level on session 1, although the regression coefficient for word span as predictor now fell short of statistical significance.

## DISCUSSION

### Problems in comprehending short reversible sentences in children with SLI

Our study found that for many children with SLI, language comprehension ability measured with TROG-E lags well behind their peer group. In addition, just over half of these children exhibited difficulty with even short and syntactically simple sentences that their age-matched peer group found extremely easy. Overall, the comprehension level of

children with SLI aged 8 to 9 years was comparable to typically-developing children who were around three years younger.

## Does repetition of sentences help or hinder learning of preposition meanings?

Children with SLI who showed poor comprehension of spatial prepositions on the initial training session had significantly higher scores on the repeated vs. the unique sentences in the second and third sessions, though they had found these sentence types equally difficult in session 1. In session 4 their performance on unique items rose to be as good as for repeated items. Together, these findings suggest a dynamic pattern of learning of syntactic constructions in these children with SLI: they first rely on repeated items in order to gain basic understanding of the meaning of specific sentence exemplars. Once they have gained some understanding of the meaning of individual sentences using these prepositions, then a more general meaning can be reliably extracted and generalised to new contexts. In contrast, no difference between repeated and unique training sentences was seen for younger grammar-matched controls until the final training session, when, unexpectedly, performance on unique items deteriorated.

Note that the pattern of performance that was observed in the children with SLI was the opposite to what might have been predicted from artificial grammar learning studies, where it has been found that variability of items facilitates learning of rule-based sequences (*Gómez, 2002*). The SLI children, but not the grammar-matched controls, were less accurate on unique than on repeated items in sessions 2 and 3. This pattern is consistent with the findings of *Hsu, Tomblin & Christiansen (2014)* who found that children with SLI did better with low-variability sequences in an artificial grammar learning study.

In our study, however, high variability (i.e., unique items) did not benefit typically-developing children either. Both groups responded faster to repeated items. To explain this, we need to consider the task demands of learning an artificial grammar compared with those of our training task. In a classic artificial grammar learning task, all that is required is for a person to demonstrate awareness of an underlying sequential structure. In our tasks, children had to assign meaning to a sequential structure. For this, we suggest, repetition may facilitate performance. Comprehension of a simple reversible sentence involves building a representation of its syntactic and semantic structure that distinguishes between the subject of the sentence and the object of the preposition, and this involves holding verbal material in a memory buffer as the structural representation is generated. There are three ways in which this process may be facilitated when items are repeated. First, it is possible that children do not analyse the sentence structure, but simply rote learn the meaning of the whole sentence: e.g., the child may learn to associate the sentence: "the apple is above the horse", with a particular spatial configuration, but not form an abstract representation of the meaning of "above". This explanation corresponds to the "rote learning" account of SLI proposed by *Hsu & Bishop (2010)*. A second possibility is that a string of words becomes easier to retain when it is repeated (Hebbian learning), and so repetition could help counteract comprehension problems due to short-term

memory limitations. In another study with these children, we found that Hebbian learning of noun sequences was less good for those with SLI than in age-matched controls, but such learning was nevertheless observed, i.e., memory was better for repeated than non-repeated sequences (*Hsu & Bishop, 2014*). A further way in which repetition may facilitate performance is by priming responses to the picture-noun combinations used in the task, so that the relevant pictures can be more rapidly selected. The child who hears "apple" many times, will know what the picture looks like and so be able to scan for and select it rapidly. This may free up cognitive capacity for computing sentence meaning. This could account for the faster responses to repeated items seen in both groups.

### Prediction of comprehension from memory measures

Children who did poorly in comprehending short reversible sentences were similar to other children in many regards but they tended to have low scores on a nonword repetition test of phonological short-term memory and on a syntax formulation task. Their comprehension scores at the end of training were significantly predicted by a measure of memory span for words, even after taking into account their initial scores in session 1. Thus ability to remember a series of unconnected words predicted how far children benefit from repeated exposure to short reversible sentences containing a spatial preposition.

At first glance, our results may seem to contradict findings by *McDonald (2008)* who found that word order was an aspect of grammar that was relatively impervious to memory effects. However, her studies focussed on detection of grammaticality violations (such as "opens the door the man"), which pose very different demands on listeners than comprehension of grammatically legal but reversible sentences.

Overall, our results are in line with the previous findings that poor short-term memory plays a role in comprehension problems in children with SLI (*Montgomery, 1995*; *Montgomery, 2000*; *Montgomery, Magimairaj & Finney, 2010*). The word span task used in the current study is similar to the training task in several aspects, and this might account for why only word span but not nonword repetition predicted the amount learned in the training task. First, in each trial of the word span task children listened to a list of familiar words (e.g., apple, star, box) instead of a single nonsense word as in the nonword repetition task. Thus the word span task tested abilities more akin to those involved in processing sentences composed of sequences of real words. In addition, in the nonword repetition task children were required to give their responses verbally, while verbal responses were not required in the word span task or the training program. This difference between the two memory tasks could also have contributed to the finding.

### Clinical implications

Results from this study offer both negative and positive messages to those concerned with intervention for comprehension problems. On the negative side, even when training was restricted to a single pair of antonyms, performance still fell below the level of age-matched controls, who made very few errors on these items. Also, neither children with SLI nor their grammar-matched controls showed improvement from pre- to post-test on TROG-E. A lack of transfer would be consistent with the idea that children tend to learn sentence

meanings by rote: such knowledge would not generalize to new meanings. Nevertheless, this latter result needs to be treated cautiously, given that there were only four relevant items in TROG-E; it is possible that a longer and more sensitive test of preposition comprehension may have revealed some improvement.It is noted that successful transfer to TROG items has been reported by *Ebbels et al. (2014)* who provided grammatical training to a group of adolescents with receptive impairments using a visual coding training strategy. The current training program differs from that used by *Ebbels et al. (2014)* in so many different ways (e.g., including training strategies, amount of training, the complexity of target structures, age of the participants, etc.) that it is difficult to know which factor(s) might account for the different findings. Nevertheless, it is reasonable to assume that any one of these factors could affect the effectiveness of intervention in clinical settings.

Results from this study also offer positive messages to those concerned with intervention for comprehension problems. First of all, children with SLI benefited from multiple exposures to the same sentence exemplars during the initial phase of intervention. This might compensate for their limitations in memory and provide the children with an opportunity to build more solid representations of the first few sentences they encountered before the general meaning of the related sentences could be detected and learned. The ultimate goal is for children to learn the general meanings of constructions such that their comprehension is not restricted to sentences they have heard before; the current results suggest that when a certain level of comprehension accuracy of repeated sentences is reached, different sentence exemplars of the same constructions could be emphasized in order to facilitate learning of general meanings. Furthermore, we found an effect of generalization from the trained prepositions to a new pair of untrained prepositions, and this effect was not limited to the younger grammar-matched children but was also observed in the children with SLI. This finding reveals positive effects of receptive grammar training that could generalize to comprehension of sentences using similar syntactic constructions. It is unclear whether or not the generalization depends on a particular training format, but our results suggest that children with SLI may do best if given repeated exposure to specific sentences with a given sentence frame, perhaps because this facilitates rapid interpretation of the nouns in the sentence, freeing up mental capacity for sentence interpretation. Accordingly, generalisation to other sentences using the same syntactic frame may be more effective if preceded by rote learning of specific items. In addition, when training children's grammatical skills, it might be beneficial to restrict consideration to a small, highly-familiar vocabulary, and not to attempt generalization to new vocabulary until performance on a restricted set of items is highly reliable. It would be of interest to conduct future studies examining how different presentations of training stimuli affect learning of a given sentence frame and subsequent generalization to a similar but different sentence type.

In conclusion, a high proportion of children with SLI had difficulty understanding reversible sentences even when these are short and syntactically simple. It is not plausible that these difficulties were due to impaired auditory perception, because the words used in the sentences were simple, discriminable and illustrated with pictures. Although some children performed at chance, others scored above chance, indicating they had some grasp

of the meanings of prepositions, but they failed to perform consistently over repeated trials. The pattern of results suggested that their comprehension failures may be related to weaknesses in verbal short-term memory. Our study suggested that performance could be improved by computerised training of sentence comprehension, incorporating rote learning of specific items in a sentence frame. Nevertheless, the training regime used in this study was not sufficient to bring performance up to the level of an age-matched peer group.

## ACKNOWLEDGEMENTS

We thank Annie Brookman, Nikki Gratton, Mervyn Hardiman, Anneka Holden, Georgina Holt and Eleanor Paine and for their invaluable assistance in data collection. Figure 2 was prepared by Tim Brock of datatodisplay.com. Finally, we are most grateful to all the schools, families and children who participated in the study.

### Funding

This study was supported by Wellcome Trust Programme Grant 053335. The funders had no role in study design, data collection and analysis, decision to publish, or preparation of the manuscript.

### Grant Disclosures

The following grant information was disclosed by the authors:
Wellcome Trust Programme Grant: 053335.

### Competing Interests

Dorothy V.M. Bishop is the author of three of the standardized assessments used to assess children in this study (CCC-2, TROG-E and ERRNI). These are published by Pearson and the royalties are paid direct to a charity.

### Author Contributions

- Hsinjen Julie Hsu conceived and designed the experiments, performed the experiments, analyzed the data, wrote the paper, prepared figures and/or tables, reviewed drafts of the paper.
- Dorothy V.M. Bishop conceived and designed the experiments, analyzed the data, wrote the paper, prepared figures and/or tables, reviewed drafts of the paper.

### Human Ethics

The following information was supplied relating to ethical approvals (i.e., approving body and any reference numbers):

University of Oxford Medical Sciences Division Interdivisional Research Ethics Committee: Reference number MSD/IDREC/2009/28.

## Data Deposition

The following information was supplied regarding the deposition of related data:

We plan to deposit the raw data in a database on the Open Science Framework once we have published one further study on a different set of variables from the same participants.

## Supplemental Information

Supplemental information for this article can be found online at http://dx.doi.org/10.7717/peerj.656#supplemental-information.

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
