# Peer review of "Training understanding of reversible sentences: a study comparing language-impaired children with age-matched and grammar-matched controls"

_PeerJ, doi:10.7717/peerj.656_

## Round 0.1 · original submission · Major Revisions

Please revise the paper considering all reviewers' comments.

·

Basic reporting

This paper presents the results of comparing children with language impairment with age-matched and grammar-matched controls on a short training that involved comprehending simple reversible sentences.

The contribution is scientifically valid. My concerns pertain to the statistical analysis, and to the fact that the language test on which children were trained was not sensitive enough, as only about half of the language impaired group scored below 90% correct. The first concern is discussed below. The second concern can be gracefully handled if the authors highlight its import in the Discussion section (both in the Abstract and in the Manuscript). In the Abstract, they use phrases such as "Many children with SLI", "children with SLI seem to do best" to discuss their results, implicitly suggesting that the findings can be readily extended to the whole SLI phenotype.

Experimental design

No comments.

Validity of the findings

The statistical analyses present with some inconsistencies:

- 310: Cramer et al. under review. It is my opinion that the reader should have all the elements to judge a paper. Therefore, referring to unpublished methods becomes acceptable only if the methods described in the unaccessible reference are succinctly but exhaustively described.

- 313: Assuming that Cramer et al. are right, and holding on to the consequential authors' stance that p <0.05 but > 0.007 should be considered marginally significant, then p = 0.084 cannot be described as marginally significant. It is nonsignificant.

- Traditionally, marginally significant (alongside nonsignificant) F statistics should be reported but not interpreted, nor attributed an effect size estimate. If this were done, the paper would increase in readability.

- 320-322: The authors followed up a significant three-way with T tests. This is, in my view, just not the correct way to go. The higher-order interaction should be broken down into lower-order (2-way) interactions, whose results then can be evaluated using T tests.

- 342: The analysis in this section goes back to include all participants. I would suggest the authors state this in the subsection title.

- 361: The interaction is marginally significant (according to the authors' own frame of reference), and thus cannot be interpreted.

- 393 ff: In the authors' opinion, could rote learning explain both the difference between repeated and unique sentences, as well as the RT advantage of SLI children? See Figure 2.

- 400: Non-word repetition was originally used as an inclusion criterion for the identification of individuals with SLI. Using it to predict one aspect of SLI (namely, the one highlighted by the training test) stands out as an illegal move, as variance of predicted values would turn out to be partially but directly determined by variance of predictor values.

- 640. Figure 2 should clearly report whether it includes results from the whole group of SLI (and controls), or only those whose performance was < 90%.

- As a reader, I would be interested in the subgroup of SLI children who performed at chance level. I am aware that the sample size is small for parametric statistics, but nonparametric options could be explored.

Reviewer 2 ·

Basic reporting

The background, study, and discussion, are presented in a clear manner.

Experimental design

The design is quite appropriate. The number of items on the final test was small but this limitation is noted by the authors.

Validity of the findings

Most of the results are entirely believable and reasonable. To lend greater credibility to one of the findings, a bit more text is needed, as described below in the general comments.

Additional comments

Comments on: ‘Training Understanding of Reversible Sentences: A Study Comparing Language-Impaired Children with Age-Matched and Grammar-Matched Controls’

This paper describes an interesting study that reports useful findings. The question of consistent versus variable sentences in teaching comprehension of above/below and before/after is one worth pursuing. The results are informative, and the authors are appropriately cautious in interpreting the results and potential clinical implications. I have three general suggestions.

First, I suggest making a more complete case for why these prepositions were selected for treatment. Are they especially emblematic of the problems of children with SLI? I suspect the authors will make reference to the earlier study showing how difficult it is to teach such things to these children. However, this does not make them the most urgent details to work on. Just a bit more justification would be helpful.

Second, and probably the most important, I would like to see some more detail on why the grammar-control children showed the unexpected drop in performance on unique items from Session 3 to Session 4. Some coverage of this issue would help because a reader might conclude that this was just a random event or sampling error and that, therefore, the seemingly interpretable results for the SLI group might also have simply been due to sampling error.

Finally, just a small detail: The kind of high variability examined in the Gómez work is somewhat different from high variability as studied here. Her artificial grammar was designed to simulate things like is X-ing in English, where the relationship between the discontinuous morphemes (is, -ing) becomes highlighted more if X shows high variability (many different verbs used for X rather than only one or a very few). The present study varied X and Y as in X is above Y in an attempt to highlight the meaning of the preposition. So this seems quite different.

·

Basic reporting

This paper discusses the difficulties children with SLI have with a particular type of reversible sentence and reports on attempts to improve their abilities in this area. The introduction also discusses the difficulties they have with other reversible sentences, but doesn’t discuss studies aiming to improve these. Given that the focus of this paper is on training, discussion of other intervention/training studies which have been carried out on these areas would be beneficial (Ebbels & van der Lely, 2001; Ebbels, 2007; Levy & Friedmann, 2009; Riches, 2013; Ebbels, Maric, Murphy & Turner 2014). In particular, Ebbels et al (2014) should be discussed as progress generalised to the TROG, unlike the current study. The possible reasons for this difference (e.g., total amount of training) could usefully be discussed.

Nonword repetition is described as a test of phonological short-term memory. However, this test relies on the integration of many other processes in addition to p.s.t.m (e.g., auditory processing, phonology, working memory and motor planning and execution), any or all of which could be impaired and lead to poor nonword repetition. This is particularly relevant because, of the two ‘memory measures’, only memory span predicted the amount learned, whereas nonword repetition did not. Perhaps this is because nonword repetition is not a pure measure of memory.

On p14 and p15, Figure 3 is referred to. I presume this should be ‘Figure 2’ as there is no Figure 3.

The comparisons between SLI and control groups on the NEPSY and ACE Syntax appear to be contradictory in Tables 1 and 3 (described on p13). It would help the reader if the authors could make clearer that this is because the data in Table 3 are for a subset of those in Table 1, thus accounting for the differing findings.

References
Ebbels, S. & van der Lely, H. 2001. Meta-syntactic therapy using visual coding for children with severe persistent SLI. International Journal of Language & Communication Disorders, 36, (supplement) 345-350
Ebbels, S.H. 2007. Teaching grammar to school-aged children with Specific Language Impairment using Shape Coding. Child Language Teaching and Therapy, 23, (1) 67-93
Ebbels, S.H., Maric, N., Murphy, A., & Turner, G. 2014. Improving comprehension in adolescents with severe receptive language impairments: a randomised control trial of intervention for coordinating conjunctions. International Journal of Language & Communication Disorders, 49, (1) 30-48
Levy, H. & Friedmann, N. 2009. Treatment of syntactic movement in syntactic SLI: A case study. First Language 29[1], 15-50.
Riches, N. 2013. Treating the passive in children with specific language impairment: A usage-based approach. Child Language Teaching and Therapy 29[2], 155-169.

Experimental design

No comments

Validity of the findings

No comments

Additional comments

Overall this is a very interesting article which is well-conducted and reported.

---

## Round 0.2 · accepted · Accept

The paper is now acceptable for publication.